# Sterilization of PLA after Fused Filament Fabrication 3D Printing: Evaluation on Inherent Sterility and the Impossibility of Autoclavation

**DOI:** 10.3390/polym15020369

**Published:** 2023-01-10

**Authors:** Jonas Neijhoft, Dirk Henrich, Andreas Kammerer, Maren Janko, Johannes Frank, Ingo Marzi

**Affiliations:** Department of Trauma, Hand and Reconstructive Surgery, Goethe University Frankfurt, University Hospital, 60326 Frankfurt, Germany

**Keywords:** 3D printing, fused filament fabrication, sterilization, autoclaving, polylactic acid, thermal deformation, traumatology

## Abstract

Three-dimensional printing, especially fused filament fabrication (FFF), offers great possibilities in (bio-)medical applications, but a major downside is the difficulty in sterilizing the produced parts. This study evaluates the questions of whether autoclaving is a possible solution for FFF-printed parts and if the printer itself could be seen as an inherent sterilization method. In a first step, an investigation was performed on the deformation of cylindrically shaped test parts after running them through the autoclaving process. Furthermore, the inherent sterility possibilities of the printing process itself were evaluated using culture medium sterility tests. It could be shown that, depending on the needed accuracy, parts down to a diameter of 5–10 mm can still be sterilized using autoclaving, while finer parts suffer from major deformations. For these, inherent sterilization of the printer itself is an option. During the printing process, over a certain contact time, heat at a higher level than that used in autoclaving is applied to the printed parts. The contact time, depending on the printing parameters, is calculated using the established formula. The results show that for stronger parts, autoclaving offers a cheap and good option for sterilization after FFF-printing. However, the inherent sterility possibilities of the printer itself can be considered, especially when printing with small layer heights for finer parts.

## 1. Introduction

Since 1978, 3D printing and additive manufacturing in its various techniques have offered the possibility and a wide variety of applications especially in medicine due to the different requirements and individualization possibilities [1]. Fused filament fabrication (FFF) impresses with its cost-effectiveness and simplicity of processing. Especially the use of multiple bioresorbable printing materials, such as polylactic acid (PLA) or polycaprolactone (PCL), adds further advantages. Trauma surgery and orthopedics, in particular, benefit from the possible applications ranging from use as bone substitute material [2,3], patient-specific implants [4,5], preoperative planning [6], use in teaching [7], or as protective equipment [8]. The requirements are as varied as the possible demands [1]. While in some cases the focus is on mechanical stability, in the case of bone substitute materials, the goal is to have as fine a structure as possible. As simple as the processing of core materials such as PLA utilizing FFF is for the previously mentioned applications, the sterilization turns out to be complicated. There are different methods:(1)Physical sterilization using gamma radiation, e.g., emitted by cobalt-60, is used in many areas of application. In addition to the medical sector, these also include the food and packaging industries. By using ionizing radiation, proteins and amino acids are destroyed, and thus germs and bacteria are killed. The disadvantages of this procedure are its high acquisition and holding costs and, during the sterilization process, possible changes in the chemical properties of thermoplastic polymers like PLA [9,10,11,12,13];(2)Examples of chemical sterilization are ethylene oxide or hydrogen peroxide plasma disinfection. It treats the surface of the product with direct alkylating agents, which destroy not only DNA/RNA but also proteins. As Oth et al. (2019) point out, chemical sterilization such as *peroxide plasma ethylene oxide* is a suitable method for genioplasty cutting guides made of PLA and PETG (polyethylene terephthalate glycol), which both suffer from deformation in the sub-millimeter range [14]. In addition, other studies name high costs, toxic hazardous residues, and in-contrast deformations on the sterilized product in short- and long-term sterilization [9,12,13,14,15,16,17,18,19];(3)One of the most frequently used methods is autoclaving. Its roots are in the 17th century, when Denis Papin experimented with his “Steam Digester” [20]. Over the centuries, with further development, it was used to kill bacteria and germs by the combination of heat, pressure, and saturated steam [21,22]. Today, there are different autoclaving protocols. A common one uses 121 °C for 20 min at 100 kPa. Due to the thermoplasticity of PLA (glass transition temperature 58–65 °C, melting temperature >184 °C) [23] and other polymers used for FFF 3D printing, autoclaving with its temperature limitations seems impossible. The effects related to the biomechanical integrity of PLA after autoclaving differ in various previous studies. For example, while Savaris et al. (2016) showed degradation and partial destruction with the formation of holes in their sterilization of PLA microfilms [19], Boursier et al. (2018) found only minor changes in the accuracy of their (bigger) models and thus considered sterilization by autoclaving to be a viable method [24].

It raised the questions of why autoclaving of PLA in some studies seems impossible due to deformation and within what limits it can be used for sterilization, and furthermore whether one can talk about inherent sterilization during the FFF printing process, as at least two components used in the autoclave, heat and pressure, also play an important role inside the extruder and print head during the printing process.

This study aimed to investigate the effects of the autoclaving process on PLA test specimens and to find out to what limits simple geometries can be autoclaved without a significant change in the initial structure and dimensions. As sterile positive controls were noticed during the investigation, the planned scope of the study was extended, and a test model was developed to examine the inherent sterility of the 3D printing process itself. 

## 2. Materials and Methods

### 2.1. Specimen Fabrication

The test specimens were designed using computer-aided design software, Fusion 360 (Autodesk, Mill Valley, CA, USA). For the sterility evaluation, simple cylinders with a length of 25 mm and a diameter of 5 mm were printed in an upright position. For the following deformation and symmetry tests, specimens with a base diameter of 10 mm, a height of 5 mm, and a total length of 25 mm were produced. These specimens were tested with ascending diameters of 1 mm, 3 mm, 5 mm, 10 mm, 15 mm, and 20 mm. Processing and generation of printable gcodes were performed using Cura (v.4.8., Ultimaker, Utrecht, The Netherlands), while printing was done on a modified fused filament fabrication (FFF) 3D printer (Ender 3, Creality 3D, Shenzen, China) with a nozzle size of 0.4 mm. Standard PLA filaments with a 1.75 mm diameter were used (DasFilament, Emskirchen, Germany). Some of the most important printing parameters are shown in Table 1. For comparability, the parameter wall line was set to 99 to ensure the printing of only the outer layers. Retraction of the filament during printing was disabled. Additional parameters were used for the inherent sterility test. These are stated in Table 2.

### 2.2. Autoclavation Process

The specimens achieved in 2.1. were then sterilized in an autoclave (Systec 2540EL, Tuttnauer, Schwabach, Germany) with the following standard process settings: first, they were heated up to a temperature of 121 °C for 20 min with 110 kPa of pressure, and they were then slowly cooled down to room temperature.

#### 2.2.1. Sterility Evaluation

A group of 10 autoclaved specimens (5 mm in diameter and 25 mm in length) and a positive control (n = 5), which was not run through the autoclaving process, were stored in RPMI standard media (RPMI 1640, Roswell Park Memorial Institute, Ref. 21875-034, Thermo Fisher Scientific Inc., Schwerte, Germany) in the incubator at 37 °C for 7 days. Media was checked daily for changes in opacity, color, or other microscopic signs of contamination.

#### 2.2.2. Deviation Measurement

For the deviation measurements, the specimens described in 2.1. were used in two different ways. Firstly, a feeler gauge (Figure 1) that did not go through the autoclaving process was used for each specimen. It was printed on the same printer with the same settings but exceeded the core radius by 0.1 mm. Starting with a test specimen with a diameter of 5 mm, they were tested to see whether they fit through the feeler gauge before and after the autoclaving process to check not only for deformation but also the preservation of symmetry (Figure 1). A replicate of 10 test specimens was used for this test setup.

The next step was to test all specimens in whose rotational orientation the deformation from the orthogonal axis was the greatest, as described in the following: A test stand was again designed using computer-aided design, into which the specimens could fit and be freely rotated. In the background, a screen of linear graph paper was used to capture the shadow of the specimen and measure the deviation from its orthogonal pre-autoclaved version. To obtain comparable data, the distance between the screen and the specimens had to be adjusted, which was achieved by using a locking screw with a ball tip and a rail mechanism. The light source and the camera for obtaining the pictures also had to be at a predefined distance from this screen to get high-resolution pictures with fixed focal lengths. The device can be seen in Figure 2. For easier handling and achieving a straight and perfectly focused optical line, the test stand was designed to fit the smartphone G8x (LG Electronics, Seoul, South Korea) in its foldable 90° standing position. In addition, a Bluetooth remote camera trigger was used to obtain 15 images of each test specimen with the angle rotated slightly to the greatest degree of deviation. An example of the obtained results can be seen in Section 3.1.2.

The pictures were then analyzed using ImageJ [25,26], and the maximum deviation was determined for each specimen. Printed but not autoclaved specimens were used as positive controls. A total of 120 specimens, 10 for each diameter, were tested.

### 2.3. Inherent Sterility

First, a clean and sterile environment had to be ensured. Inside a workbench with a hood and continuous airflow (Safe 2020, Thermo Fisher Scientific Inc., Schwerte, Germany), a smaller 3D printer (modified Ender-2, Creality 3D, Shenzhen, China), which shares the same extruder and print head as the Ender 3, was used. The heated bed surface was exchanged for a glass-reinforced epoxy laminate (FR-4) build plate, for easier disinfection. After covering the electronics, the whole printer was first precleaned with a 96%vol alcoholic solution and then placed under the hood of the workbench. Afterward, the printer was exposed to UV light for 30 min to ensure a germ-free printing environment. Neither the filament nor the specimens were treated. After loading it into the printer, the test specimens were printed with two different printing settings. The first batch of test specimens (n = 10) was printed using the same settings as mentioned in Table 1, which experience has shown to be a good foundation and was used in previous manuscripts [7]. These will be described as “trial stage 1” from now on. For the comparison group, i.e., trial stage 2, a higher temperature and longer thermal contact time of the PLA in the print head were aimed for. Therefore, the extrusion temperature was set to higher values with the highest possible cooling fan speeds to compensate for the increased viscosity. To maximize the time in the print head, the printing speed was slowed down and the line width was reduced. The printing setup can be seen in Figure 3. The printed parts were then transferred with sterile forceps into a 6-well plate (Sarstedt, Nümbrecht, Germany) containing RPMI media. Afterward, they were stored for 7 days in an incubator at 37 °C and checked every day for indirect signs of contamination, like changes in color or transparency. As a positive control, non-printed and non-treated filament were incubated with the same protocol.

### 2.4. Statistical Analysis

Measurement data in this work are presented as means with standard deviations of the mean (SD), unless noted otherwise. Results were considered significant for values of *p* < 0.05. The values are graphically displayed, while statistical analysis was performed with IBM SPSS Statistics 28 (IBM, New York, NY, USA).

### 2.5. Ethics

There are no ethical concerns regarding this study, for this reason, no ethics vote was needed.

## 3. Results

### 3.1. Specimen Fabrication

The test specimens were printed from PLA and were easily obtainable with both the printing settings as stated in Table 1 and Table 2.

#### 3.1.1. Autoclavation Process and Evaluation

Autoclaving, as one of the main methods with wide application to achieve sterility, also showed effectiveness in this test, with no signs of contamination in the medium containing the test specimens or RPMI media. The geometric changes suffered by the test parts will be discussed in the next section.

Interestingly, even the positive controls continued to show sterility after the printing process in two out of five cases. So only three samples developed a visible color change on day 3 and then streaking inside the medium the day after, so these samples were evaluated as contaminated.

#### 3.1.2. Deviation Measurement

Deviation measurements were fulfilled using the previously described test stand.

Here, as expected, the test bodies became less susceptible to deviations with increasing diameter. The bending curve of the test bodies was similar optically, so there could always be a deviation of the axis in one direction with a maximum value. The geometry deviations obtained were largest for the 1 mm test body diameter, with a mean of 12.5 mm (SD 4.6) axis deviation. This was followed by the group with a 3 mm diameter and a mean deviation of 1.5 mm (SD 0.7). These two groups could already be macroscopically identified as deformed. The mean axis deviation of the 5 mm measuring test group was 0.2 mm (SD 0.2), and that of the 10 mm group was 0.1 mm (SD 0.2). The test bodies with 15 mm and 20 mm no longer showed such detectable deviations. The non-autoclaved test specimens were all completely orthogonal and without deviation. An example of the test series is seen in Figure 4.

Regarding the preservation of symmetry, the picture was similar (see Figure 5). At 5 mm, five out of ten of the specimens were still symmetrical, while in the group with a 10 mm diameter, nine out of ten showed preserved symmetry, within the limits of the feeling gauge. From 10 mm onward, i.e., in the 15 and 20 mm groups, symmetry was preserved after autoclaving. The positive controls, which were not autoclaved before the test, could all be pushed through the gauge.

### 3.2. Inherent Sterility

The positive control experiments with the autoclaving process gave rise to the idea of investigating the possible inherent sterility of the 3D printing process itself. As already described in the Methods Section, only the environment and the printer itself were disinfected. The filament was not sterilized but used directly in the printer. The printing parameters for trial stages one and two are described in Table 1 and Table 2. Using PLA, which is easily printable, both trial stages showed very good and consistent results. The specimens could be easily detached from the print bed and transferred with sterile forceps to the 6-well plate without any problems.

The untreated filament samples, which served as a positive control, showed incipient changes in the medium in the sense of a color change after only 2 (n = 7) or 3 days (n = 3), and signs of streaking on the following day. Thus, the contamination of all samples could be determined. The result of the trial stage 1 specimens was different: only one out of ten specimens showed a color change on day 6. The test specimens from trial stage two, which were printed at a higher printing temperature and at slower printing speeds, showed no evidence of contamination over the entire observation period.

## 4. Discussion

Fused filament fabrication offers many possibilities, especially in terms of rapid prototyping and micro- and macrostructures that are cheap and easily achievable. However, since many application areas require the absence of bacteria and germs, the parts must be further processed to achieve sterility. Sterilization of 3D prints made of thermoplastic can be a difficult task: on the one hand, these materials are easy to process, but on the other, their susceptibility to heat or radiation [10,11,17,23,27] seems to limit the sterilization possibilities. As already mentioned in the introduction, each of the different sterilization types has its disadvantages. Probably the most common type is autoclaving, but since thermoplastics tend to deform under heat and pressure, it is always claimed that autoclavation is impossible. To refute this statement, this study shows some boundaries within which autoclaving is possible.

It must be stated that for this study, new PLA (DasFilament, Emskirchen, Germany) was used, which was not overstored or humidified over time and underwent the printing process for the first time. These changes can be seen and heard during the printing process. For example, changes in viscosity, bubbles at the tip of the nozzle, or a crackling noise during printing can indicate overstored or wet filament and have major effects on the outcome of the printed parts. It is conceivable that remaining humidity in the printed parts can also result in worse results during the autoclaving process with poor inter-layer adhesion. [28,29]

### 4.1. Autoclaving of FFF-Printed Parts

As presented, deformation occurs mainly in small parts and is therefore not an option for the production of finely structured objects with their microporosity and details down to 100 µm [2], as these would be destroyed. However, our results showed that parts with fewer details and stronger structures, such as those used for laboratory supplies, can withstand the autoclaving process without major deformation. Although it was not tested explicitly, our experiences and different studies, such as Rozema et al. (1991), suggest that the mechanical properties of FFF-printed parts made from PLA can be strengthened by heat treatment by improving inter-layer adhesion further [13,30,31,32]. During the printing process, each layer bonds to the layers around it, increasing the structural strength of the entire part. This bond is increased by higher temperatures during printing or heat treatment within certain temperature borders. Contrary to that, small voids and gaps between the layers, resulting from shrinkage while cooling and remaining heated moisture inside the filament, weaken the part by lowering inter-layer adhesion. Taking this into account, and if not overdone, autoclaving could result in not only sterile but also stronger parts as layer adhesions improve [13,27,32,33].

The symmetry tests revealed equal results, showing not only that the 3D printer prints within the specified tolerances but also that shrinkage as well as the extension of PLA during autoclavation can be neglected for bigger and stronger constructions. Depending on the accuracy and the accepted possible deformation, parts down to the smallest diameter of 5–10 mm can still be sterilized by autoclaving. It must be added that filaments from brands other than the one used in this study may perform slightly differently, so the absolute values may differ. If the biodegradability or printability of PLA is not required, thermoplastics that are printed at higher temperatures due to their higher glass transition and melting temperatures could yield even better results.

### 4.2. Inherent Sterilization

During processing, two influences mainly act on the filament within the print head: Pressure and temperature. The pressure depends on multiple parameters, such as print speed, temperature, and type of filament, but also on predefined factors such as nozzle diameter and type of extrusion system. Temperature, as the suspected main effect of inherent sterilization, can be easily measured using an infrared thermometer and adjusted or corrected by software. The contact time at which the filament is exposed to that temperature is more difficult to determine. It also depends directly on several print parameters: including layer height, line width, print speed, heat zone length, and more mundane parameters such as filament diameter.

To obtain an approximate determination of the contact time, a multi-colored filament was first produced by splicing different filament colors together (first white, then red, then black). The first admixture of red particles in the initially white print object was evaluated as the zero point. As soon as black particles appeared in the printed object, the time was stopped, and the experimental contact time inside the print head could be approximated. However, since this is only an approximate determination, it should also be determined mathematically. The parameters influencing the contact time were summarized as shown in Equation (1).
(1)$$t\left(f\right)=\frac{(l_{Hb}+l_{Nl})*{\left(\frac{d_{F}}{2}\right)}^{2}}{\frac{d_{lw}}{2}*\frac{d_{lh}}{2}*f}$$
where:*d_lh_*—layer height;*d_lw_*—line width;*f*—max. print speed;*l_Hb_*—heat block length;*l_Nl_*—nozzle length;*d_F_*—filam. diam.

Using this equation, the contact time can be calculated as 44.02 s for the first trial stage at a printing speed of 40 mm/s. The settings used for the second trial stage (see Table 2) resulted in a longer contact time of 117.39 s. The calculation approximates a linear printing process and does not consider any special slowdowns caused by design geometries or process-specific changes. In reality, the speed is likely to be slower, since the actual flow speed is still somewhat reduced during, e.g., the outer layer, empty travel path movements of the print head, or filament retraction. Furthermore, the contact time can be greatly increased by structural changes such as a longer heat zone or by finer as well as slower prints.

Considering the results, the inherent sterilization process of the printer itself can be seen as a sterilization option. The obvious benefit here is that no additional machine or work step is needed. This could be of particular interest to laboratories that do not have access to expensive equipment such as gamma radiation or ethylene oxide chambers. There are some predominant advantages to inherent sterilization when printing fine parts, as they require certain changes in printing parameters. In particular, the printing speed needs to be slowed down, and temperatures may need to be raised due to the use of composite materials [34]. According to Equation (1), utilization of smaller nozzle sizes with the resulting smaller layer widths and finer layer heights will raise the heated dwell time inside the extruder further.

In our study, we used PLA because this polymer has been used in previous studies [3,34] because of its biocompatibility [35,36] and printability. However, printing with polymers such as glycol-modified polyethylene terephthalate (PETG) or polyamide can further improve the inherent sterilization results of the printing process through higher printing temperatures.

Of course, this technique should not be used for its intended use in patients, as there is no deeper evaluation of the process and its possibility to kill all germs or even deactivate spores. However, some major advantages of 3D printing—its high flexibility and fast individual part production [2]—could even be extended, as sterilization is easily achievable for initial pilot projects, proofs-of-concept in cell cultures, and geometry comparisons. Although not initially intended for this study, our results suggest basic sterility due to the printing process itself. Postulating that, one could even go one step further and investigate what is the lowest needed “dwell time to kill” regarding material-specific printing temperatures. Knowing that specific constant, one could force the slicer to preserve sterility by adjusting the minimum print speed, material flow, or resting times of the extruder while maintaining the heated filament inside. It is possible that the development of a different extruder geometry will result in a specific improved process: elongation of the heat break or nozzle length will increase the heated time of the filament. However, it must be considered that heating polymers above a certain temperature will accelerate the degradation of the filament, resulting in unusable and brittle parts as this fundamentally changes the material’s properties [23,37]. In 1988, Jamshidi et al. discovered that PLA degradation via thermohydrolysis and depolymerization with breakage into oligomers is accelerated above 190 °C, with exposure time increasing with temperature [23]. In FFF printing applications, these temperatures (or slightly below) are needed for the printing process. In this study, we examined two different sets of printing parameters (Table 1 and Table 2), which can be considered a foundation, but in future studies, it certainly must be evaluated further to determine at what boundaries inherent sterilization printing can be used for sufficient sterility for each polymer or if severe degradation occurs earlier.

## 5. Conclusions

In the present study, the influence of simple sterilization processes on the dimensional stability and sterility of 3D printed structures based on PLA was analyzed. Autoclaving while maintaining dimensional stability is only possible for coarse structures. Thus, the geometry of the printed parts is essential when choosing autoclaving for sterilization, as in the case of PLA specimens, diameters above 5 mm showed enough structural strength to withstand the application of heat, pressure, and steam during the autoclaving process. These findings confirm that autoclaving can be considered, e.g., for anatomical models or larger laboratory devices.

As finer objects such as those used in tissue engineering applications suffer from coarse deformation, inherent sterilization can be seen as an alternative: the process of using the print head for sterilization depends on adjustable parameters like layer height and line width and geometrical factors like nozzle length, heat block length, and filament diameter. The resulting temperature contact time can be calculated using the given formula. The extent to which this type of sterilization is sufficient for critical applications in vitro and in vivo must be evaluated in further studies.

## Figures and Tables

**Figure 1 polymers-15-00369-f001:**
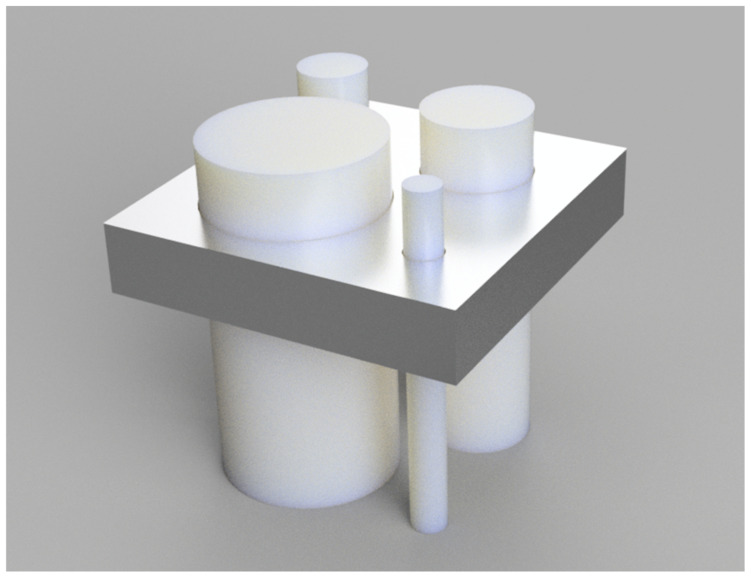
A feeler gauge was designed for symmetry preservation checks. Starting with 5 mm, the test specimens had to fit through the gauge before and after the printing process with a radius tolerance of 0.1 mm.

**Figure 2 polymers-15-00369-f002:**
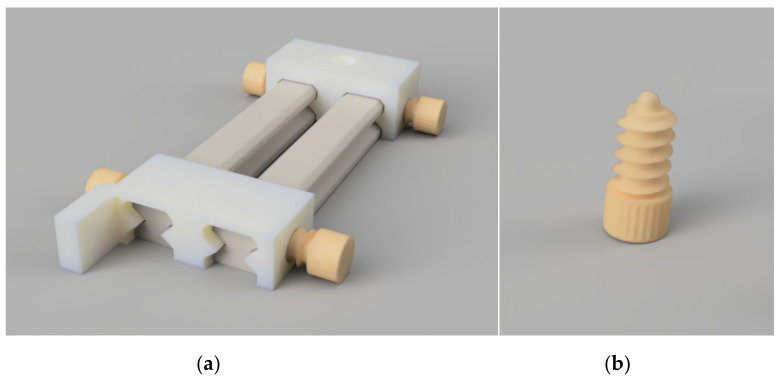
With the above-shown test stand (**a**), the camera position, specimen rotation, and distance to the screen could be adjusted and locked with ball-tip locking screws (**b**).

**Figure 3 polymers-15-00369-f003:**
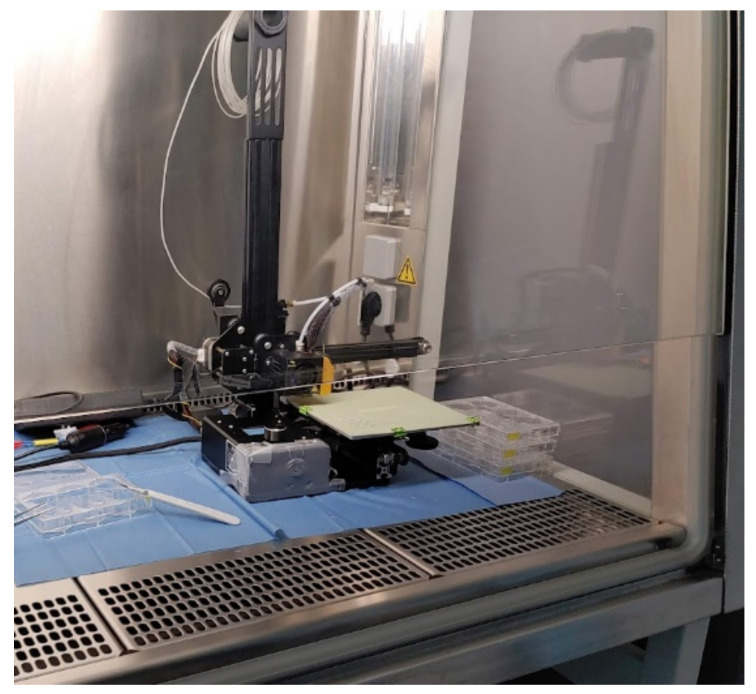
The printing setup for inherent sterility evaluation in sterile surroundings under the hood of a workbench.

**Figure 4 polymers-15-00369-f004:**
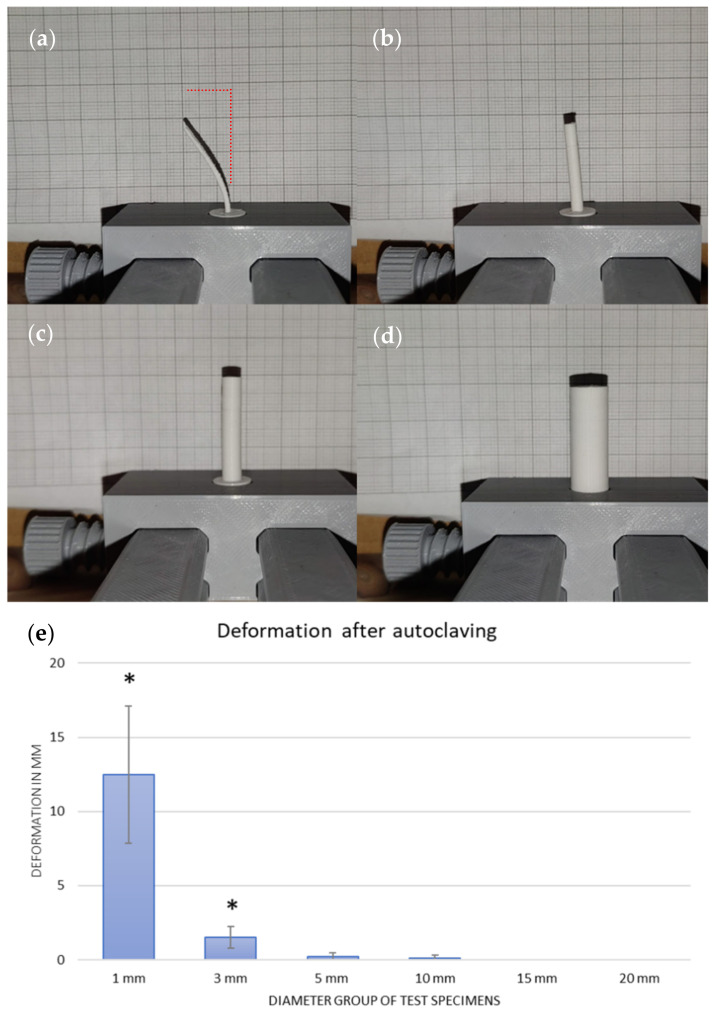
The reduction of the deviation with ascending diameter of the specimen can be seen. The specimens are rotated to their maximum degree of deviation, starting with 1 mm (**a**), 3 mm (**b**), 5 mm (**c**), and 10 mm (**d**). (**e**) A diagram which shows the resulting biggest deformation of one axis of the test specimen in mm and the related SD. After autoclaving, the groups with diameters ranging from 1 to 3 mm showed significant deformation (shown by “*”).

**Figure 5 polymers-15-00369-f005:**
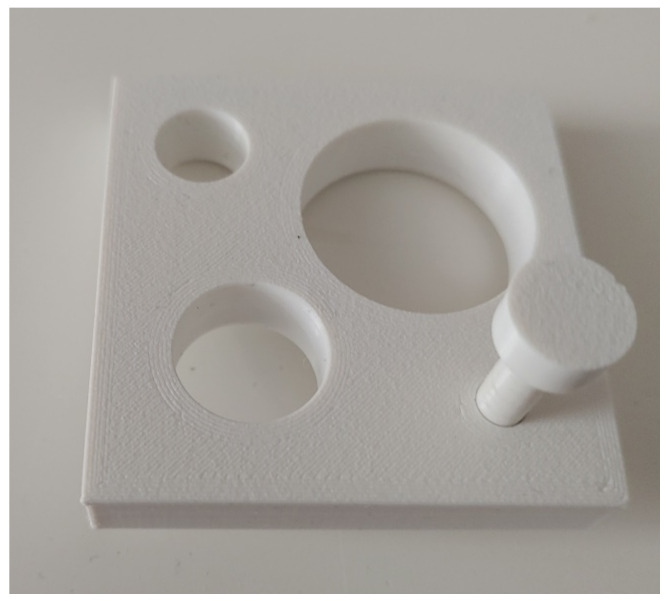
The symmetry test of a specimen with a 5 mm diameter. It could still be easily pushed through the gauge after the autoclaving process.

**Table 1 polymers-15-00369-t001:** For the printing of the specimens, the following parameters were used. It shows the global standard settings used for most printed parts of this study, except the second printing stage of the inherent sterility test, which are stated in Table 2.

Nozzle size	400 µm
Layer height	100 µm
Line width	400 µm
Outer wall speed	40 mm/s
Inner wall speed	40 mm/s
Printing temperature	205 °C
Wall line count	99

**Table 2 polymers-15-00369-t002:** For the second printing stage and inherent sterility tests the parameters shown in Table 2 were used. However, the optimal parameters depend on different printers and filaments, which always need to be adjusted individually.

Layer height	100 µm
Line width	300 µm
Outer wall speed	20 mm/s
Inner wall speed	20 mm/s
Printing temperature	225 °C

## Data Availability

Not applicable.

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
