# Peer review of "Sterilization of PLA after Fused Filament Fabrication 3D Printing: Evaluation on Inherent Sterility and the Impossibility of Autoclavation"

_polymers, 2023, doi:10.3390/polym15020369_

Round 1

Reviewer 1 Report

General observations:

The authors must improve the introduction and include more related previous research in the introduction. The previous works in this specific domain are not clearly mentioned. Furthermore, the novelty of this research compared to previous research must be indicated. There are several studies similar to this topic. such as:

https://doi.org/10.1016/j.msec.2016.07.031

https://doi.org/10.2174/1874210601913010410

https://doi.org/10.1177/2041731416648810

https://doi.org/10.3390/polym13132115

I highly invite the authors to clarify the innovation of the article.

The authors didn't provide any information about the degradation of the PLA when it is exposed to UV light. Considering during the inherent test the samples are exposed to UV the authors must explain in detail its effect on the structure and properties. It could be done by performing more experimental studies or through a literature review.

This study is about the sterilization of the 3D printed part by FFF process. However, there is no in-situ, in-vivo test on the samples to measure if the printed parts are sterilized.

In the inherent sterilization section, the only method to measure the efficiency is color change of the printed sample. This is the weak point.

Please provide more information about the degradation of the PLA in autoclave condition and UV.

A detail of the review could be found in the attached file.

Reviewer 2 Report

The authors deal with the efficiency of the sterilization of PLA after Fused-Filament-Fabrication 3D printing and the dimensional stability with the treatment of autoclavation. The main idea is interesting but the manuscript should be largely improved.  The suggestions listed are recommended to be considered. 

1. The abstract should be reorganized as a complete paragraph. Do not give brief descriptions of each section. The background can be just a single sentence, and the destination as well as the results should be emphasized. To tell the readers what you want to do and what you have done. 

2. The description of "Deviation measurement " is not clear, especially in Fig. 2. 

3. The experiment design should be scientific and presented in detail. The authors just demonstrated two key-point evaluations. This is not enough to explore the effect of autoclavation. Readers are just told it is effective, but the dependence is confusion. 

4. Personally, I'd like to place the discussion section just with the results section together. Normally, we illustrate the results, followed by the discussion.

5. " in future studies it certainly must be evaluated in which boundaries inherent sterilization printing can be used for sufficient sterility for each polymer or if severe degradation occurs earlier." This is interesting, but there is a lack of investigation. I suggest supplementing this data in the revision of a resubmission. 

6. The conclusion is too simple. The whole manuscript looks like an experimental report, not a scientific paper. Please revise it carefully. 

7. Last but not least, the whole manuscript should be re-constructed. 

Round 2

Reviewer 2 Report

It seems that this manuscript was revised according to the suggestions, and it may be considered to be accepted with a revision. But one important point which is missing is the lack of scientific highlights. The authors should pay more attention to this critical issue. A revision is further recommended to make on this manuscript.